# Impact on Sensory and Aromatic Profile of Low Ethanol Malbec Wines Fermented by Sequential Culture of *Hanseniaspora uvarum* and *Saccharomyces cerevisiae* Native Yeasts

**María Victoria Mestre** [1,2,†], **Yolanda Paola Maturano** [1,2,*,†], **Candelaria Gallardo** [1,2], **Mariana Combina** [2,3], **Laura Mercado** [3], **María Eugenia Toro** [1], **Francisco Carrau** [4], **Fabio Vazquez** [1] and **Eduardo Dellacassa** [4]

1   Instituto de Biotecnología, UNSJ, Av. San Martín 1109 (O), San Juan 5400, Argentina
2   Consejo Nacional de Investigaciones Científicas y Tecnológicas (CONICET), Godoy Cruz 2290 C1425FQB CABA, Argentina
3   Estación Experimental Agropecuaria Luján de Cuyo (INTA), Av. San Martin 3853 Luján de Cuyo, Mendoza M5507, Argentina
4   Laboratorio de Biotecnología de Aromas, Facultad de Química, UdelaR. Av. Gral. Flores 2124, Montevideo 11800, Uruguay
*   Correspondence: paolamaturano@yahoo.com.ar Tel.: +54-0264-4211700/+54-0264-4619625
†   These authors contributed equally.

**Abstract:** It is well known that high ethanol levels in wines adversely affect the perception of new wine consumers. Moreover, numerous issues, such as civil restrictions, health risk and trade barriers, are associated with high ethanol concentrations. Several strategies have been proposed to produce wines with lower alcoholic content, one simple and inexpensive approach being the use of new wine native yeasts with less efficiency in sugar to ethanol conversion. Nevertheless, it is also necessary that these yeasts do not impair the quality of wine. In this work, we tested the effect of sequential culture between *Hanseniaspora uvarum* BHu9 and *Saccharomyces cerevisiae* BSc114 on ethanol production. Then, the wines produced were analyzed by GC-MS and tested by a sensorial panel. Co-culture had a positive impact on ethanol reduction and sensory profile when compared to the *S. cerevisiae* monoculture. Wines with lower alcohol content were related to fruity aroma; moreover, color intensity was associated. The wines obtained with *S. cerevisiae* BSc114 in pure conditions were described by parameters linked with high ethanol levels, such as hotness and astringency. Moreover, floral profile was related to this treatment. Based on these findings, this work provides a contribution to answer the current consumers' preferences and addresses the main challenges faced by the enological industry.

**Keywords:** low-ethanol wines; sequential culture; *Hanseniaspora uvarum* yeast; aromatic/sensorial profiles

## 1. Introduction

Well-structured and full-body wines have become the preferences of many new wine consumers. In order to obtain these characteristics, it is necessary to ensure optimal phenolic maturity of grapes, which requires longer grape ripening times [1]. However, in the context of global warming, this practice results in a significant increase in the berry sugar content at the moment of harvesting, and consequently higher alcohol levels in the wine [2]. Numerous issues are associated with high ethanol levels in wine such as consumers' rejection, civil restrictions, health risk, and trade barriers [1,3,4]. The sensorial

quality of wines is also significantly affected because of an increase in the perception of bitterness, sweetness, astringency and hotness, and masking of volatile aromatic compounds [5,6]. In this context, different technological solutions have been evaluated: harvest of unripe berries, increase in crop load, shading bunches, choosing proper irrigation techniques, and modulation of source–sink relationships by removing leaves or topping shoots [7–10]. Other authors have tried partial dealcoholization with physical methods [11–13].

More recently, microbiological solutions have been proposed by using selected non-*Saccharomyces* and *Saccharomyces cerevisiae* yeast strains in simultaneous or sequential fermentations [4,14–16]. The use of non-*Saccharomyces* yeasts has become a common trend in the main wine regions, particularly because of their effects on the composition, flavor and color of the wine [17,18]. In addition to the aforementioned effects, this yeast group is also known to be less efficient in the production of ethanol from consumed sugars when compared with *S. cerevisiae* yeasts [19].

*Hanseniaspora* genera as a whole and particularly *Hanseniaspora uvarum* species are non-*Saccharomyces* yeasts commonly encountered at high concentrations on the grape surface and throughout the fermentation process [20]. Recently, 28 *H. uvarum* isolates were evaluated by our research group and they demonstrated interesting enological characteristics such us: ability to grow at high sugar, ethanol and $SO_2$ contents; to produce high concentrations of glycerol; low acetic acid and hydrogen sulfide levels; and the release of proteolytic enzymes [21]. Moreover, it is important to highlight that *H. uvarum* was also found to be a potential candidate to produce less ethanol because it requires more than 19 g/L of consumed sugar to produce 1% *v/v* of ethanol [21]. In a more recent study, a selected *H. uvarum* yeast strain was assessed in sequential inoculations with *S. cerevisiae* yeasts under optimized fermentation conditions [22]. The authors found that the ethanol levels were significantly reduced compared with fermentations carried out with *S. cerevisiae* monocultures. Nevertheless, and in order to achieve holistic knowledge, the aim of the present work was to assess the aromatic impact of an optimized inoculum of *H. uvarum/S. cerevisiae* yeasts in fresh must and compare the findings with a *S. cerevisiae* monoculture. It is also relevant to establish how ethanol reduction affects sensorial perception. The results would allow the design of a comprehensive microbiological strategy in order to answer the current consumers' preferences and address the main challenges faced by the enological industry.

## 2. Materials and Methods

### 2.1. Microorganisms

*Hanseniaspora uvarum* BHu9 and *Saccharomyces cerevisiae* BSc114 were used in the present study. Both strains of yeasts were previously selected based on their oxidative and fermentative metabolism in order to obtain reduced ethanol wines [21]. Strains were obtained from the Culture Collection of Autochthonous Microorganisms (Institute of Biotechnology, School of Engineering—UNSJ, San Juan, Argentina) and preserved at −80 °C until use.

### 2.2. Yeast Inoculum Preparation

Each strain was grown on YEPD agar for 48 h and the biomasses were transferred to YEPD broth (130 rpm during 4 h) [22]. Then, strains were transferred to grape must (13° Brix, pH 3.8) supplemented with 0.1% yeast extract and 0.4% peptone, and incubated at 25 °C during 24 h under aerobic conditions (130 rpm). YEPD broth was used for pre-adaptation in order to reduce the lag-stage in the grape must, which allows strains to grow immediately exponentially in the grape juice [22]. Once the pre-adaptation process had finished, cells were counted with an improved Neubauer chamber.

### 2.3. Grapes and Vineyard Location

All experiments were carried out using *Vitis vinifera* L. cv. Malbec grapes harvested during the 2017 vintage from a vineyard located in Cañada Honda, San Juan, Argentina (31°58′34′′S 68°32′52′′W)

at 610 m altitude. Grapes were manually destemmed and mixed to obtain a homogeneous solution. The composition of the fresh juice was as follows: sugar (glucose and fructose), 238.2 g/L; pH, 3.8; titratable acidity, 5.3 g/L; and yeast assimilable nitrogen (YAN), 175 mg/L. Then, 5-L vessels equipped with a Muller valve were filled with juice (3 L) and supplemented with 50 ppm of free $SO_2$ before fermentation. The Muller valve was filled with a solution of 50% sulfuric acid and 50% sterile water distilled. Vinifications were performed in triplicate.

## 2.4. Inoculation and Winemaking

Lab-scale fermentations were conducted under optimized factors previously determined by Maturano et al. [22]. Treatment 1 (T1): *H. uvarum* BHu9 was inoculated (T0) at a concentration of $5 \times 10^6$ cells/mL, and 48 h later, $2 \times 10^6$ cells/mL of *S. cerevisiae* BSc114 were sequentially inoculated. In parallel, a single culture of $2 \times 10^6$ cells/mL of *S. cerevisiae* yeasts was inoculated at T0 as control treatment (TC). Both fermentations were performed at $25 \pm 1$ °C under static conditions. Musts were supplemented with nitrogen by adding 20 mg/L of $(NH_4)_2HPO_4$ twice: after 48 h and in the middle of the fermentation (when 5% weight loss was verified). Nitrogen supplement was established based on nitrogen uptake previously determined with selected yeasts (data not shown). Punch down was carried out every 24 h in order to keep acceptable dissolved oxygen levels throughout the process. The fermentation progress was evaluated by the weight loss caused by CO2 production and vessels were weighed every 24 h.

Samples were collected periodically and viable cell counts were determined by plating onto Wallerstein Laboratory Nutrient (WLN) Agar medium (Oxoid, Hampshire, UK). Dilutions of $10^{-3}$, $10^{-4}$ and $10^{-5}$ were spread onto WLN agar medium and incubated for 7 days at 28 °C. Green colonies (*H. uvarum* BHu9) and creamy colonies (*S. cerevisiae* BSc114) were differentiated and counted [23].

After the sugar was completely consumed, 50 mg/L of free $SO_2$ was added. The wines were chemically stabilized, filtered, bottled, and conserved at $16 \pm 1$ °C until sensorial analysis. Samples of 50 mL were stored at $-20$ °C in order to carry out volatile composition analysis.

## 2.5. Chemical Analysis

Glycerol, residual sugars, total acidity and acetic, malic, lactic, and tartaric acid were measured periodically using an ALPHA FT-IR Wine Analyzer (Bruker Optik Gmbh, Ettlingen, Germany). Ethanol concentration was determined according to the OIV OENO 379-2009 ES official method. The pH was measured with a multi-parameter Adwa (AD1030 PHM_MES_6362).

## 2.6. Sensorial Analysis

After 4 months of bottle stabilization, wines were evaluated by descriptive analysis according to Lawless and Heymann [24]. A well-trained panel carried out the evaluation of 13 sensorial attributes: three color/appearance descriptors (color intensity, red and brown color), five aroma descriptors (mineral note, frutal, floral, chili pepper, and toasted) and five taste parameters (acidity, sweetness, astringency, hotness, and bitterness). The intensity of each attribute was assessed using a structured scale from 0 to 5, where 0 indicates that the descriptor was not perceived and values between 1 and 5 indicate that the intensity of the descriptors was very low to very high. The panel consisted of seven individuals (five males and two females between 35 and 50 years old) from the Wine Sensorial Analysis Department (Instituto Nacional de Vitivinicultura, Mendoza, Argentina). Vinifications were tasted blindly and in duplicate from a constant volume of 30 mL at room temperature.

## 2.7. Free aromatic Analyses

### 2.7.1. Solid Phase Extraction (SPE)

The extraction of aroma compounds was performed by adsorption and the molecules were separate elutions from an Isolute ENV+ cartridge (IST Ltd., Mid Glamorgan, UK) packed with 1 g of

the highly cross-linked styrene divinylbenzene (SDVB) polymer according to Boido et al. [25] with some modifications.

### 2.7.2. GC-MS Analyses

GC-MS analyses were conducted using a Shimadzu QP 2020 (Shimadzu Corporation, Kyoto, Japan) mass spectrometer. A Carbowax 20 M capillary column (Agilent Technologies, Walt and Jennings Scientific, Wilmington, DE, USA) (30m × 0.25mm × 0.25μm film thickness) was used. The experimental conditions were as follows: The initial column temperature was 40 °C (8 min), which was then increased to 180 °C (3 °C/min) and then increased again to 250 °C (20 min) at 20 °C/min; injector temperature, 250 °C; injection mode, split; split ratio, 1:30; volume injected, 1.0 μL; carrier gas H2, 30 kPa; energy 70 eV. The wine aroma components were identified by comparison of their linear retention indices (LRI) determined with a homologous series of n-alkanes (C9–C26), with those from pure standards or reported in the literature according to their elution order with Carbowax 20 M [26–28]. Comparison of mass spectral fragmentation patterns with those stored in databases was also performed. GC-MS instrumental procedures using 1-heptanol as an internal standard were applied for quantitative purposes. GC-MS analyses were carried out with two samples of each wine.

### 2.8. Odor Activity Value (OAV) and Relative Odor Contributions (ROCs)

The contribution of each volatile compound was quantitatively evaluated using Odor Activity Values (OAVs). The OAV was obtained by dividing the mean concentration of each volatile compound by its odor threshold value in a hydroalcoholic solution [29]. The volatile compounds contribute to wine aroma when its concentration in wine is above the perception threshold, therefore, the OAV value is above 1. In this study, the threshold values were obtained from information available in the literature. Moreover, the identified compounds were classified according to aromatic descriptors and grouped in seven aromatic series which were classified according to the associated descriptor: 1, solvent; 2, sweet; 3, herbaceous; 4, floral; 5, fruity; 6, fatty; and 7, toasted.

From the volatile compounds that presented OAV > 1, the relative odor contribution (ROC) was calculated. The relative odor contribution (ROC) represents the percentage of contribution of a particular aroma compound and this was determined as the ratio between the OAV of the respective compound and the total OAV of each wine ((individual OAV/$\sum$OAV) * 100) [30].

### 2.9. Statistical Analysis

Chemical data and population analysis were expressed as the means ± standard deviation from three repetitions and aromatic analysis as the means of two repetitions. One-way ANOVA was used to evaluate differences between treatments. Statistical analysis was performed using the InfoStat professional version (Cordoba, Argentina, 2016).

## 3. Results

The current study assessed the contribution of *H. uvarum* BHu9 and *S. cerevisiae* BSc114 yeasts to the ethanol content and sensorial and aromatic impact on wine.

### 3.1. Fermentative Kinetics and Population Dynamics

In the present study, fermentative kinetics are represented by sugar consumption and CO2 release in both fermentations: BHu9/BSc114 (T1) and BSc114 (TC) (Figure 1). Both treatments completed alcoholic fermentation after 8 days. During the first 24 h, both vinifications showed a similar sugar consumption, but from day 2 until day 6, T1 exhibited a slower fermentation rate than TC ($p < 0.05$). During day 7 and 8, sugar consumption was not significantly different ($p > 0.05$), and at the end of the process, both treatments behaved similarly.

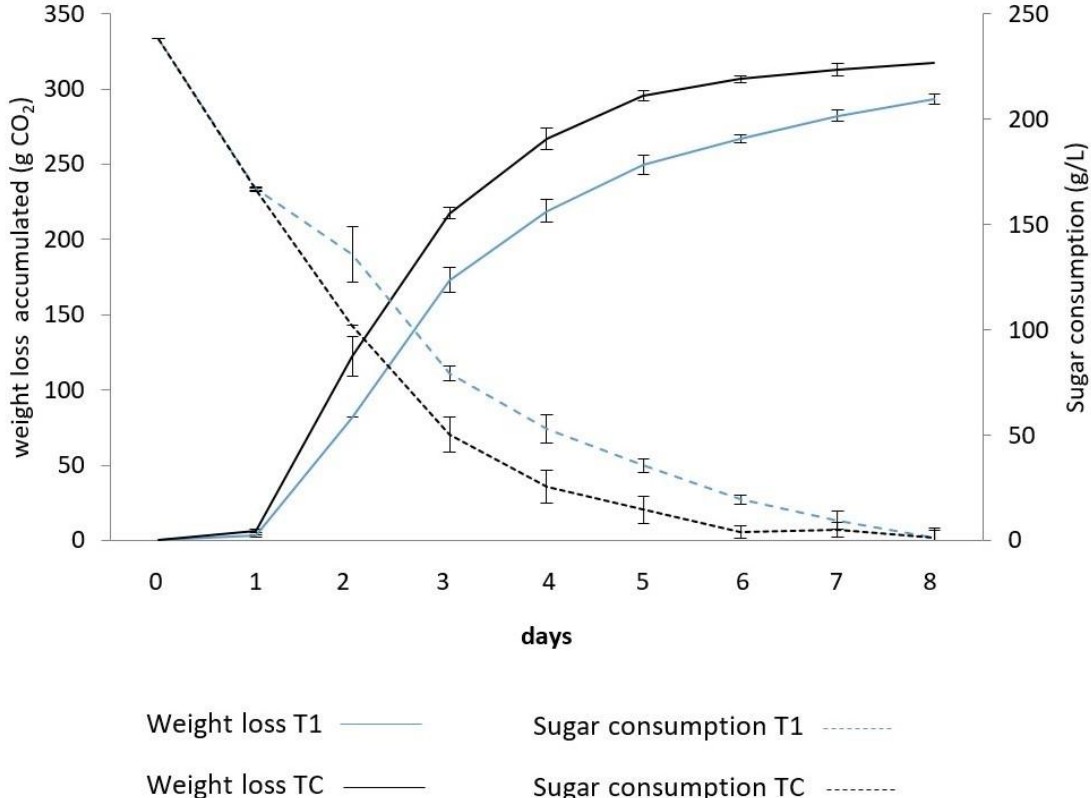

**Figure 1.** Release of CO2 (g) and sugar consumption in T1 (BHu9/BSc114) and TC (BSc114 control).

Like the sugar consumption, CO2 release showed a similar behavior for both treatments during the first 24 h. From day 2 until the end, T1 demonstrated a lower rate than TC (BSc114) ($p < 0.05$). Total CO2 production was 293.33 ± 17 g and 320 ± 10 g for the BHu9/BSc114 co-culture and **S. cerevisiae** monoculture, respectively (Figure 1).

The population dynamics of T1 (*H. uvarum* BHu9/*S. cerevisiae* BSc114) and TC (*S. cerevisiae* BSc114) are shown in Figure 2. *H. uvarum* BHu9 population increased during the early stages reaching a maximum of $8.18 \times 10^7$ cells/mL on day three. During the first 48 h, (previous *S. cerevisiae* inoculation) BHu9 consumed 102.54 g/L of sugar with an ethanol production of 3.49% *v/v*. Therefore, when BSc114 was inoculated (after 48 h), the available sugar concentration was 135 g/L. In co-inoculation trials, *H. uvarum* BHu9 maintained its population up to day 4, after which the concentrations were undetectable with the technique applied in this study. Hence, *H. uvarum* BHu9 and *S. cerevisiae* BSc114 coexisted only during 2 days. During this coexistence period, the sugar consumption was 51.74 g/L, which means that the sugar consumption by both strains was less than the consumption by BHu9 before *S. cerevisiae* inoculation, and the ethanol production at this stage was 5.93% *v/v*. At the final fermentation stage (day 5 to 8), *S. cerevisiae* BSc114 consumed 74.67 g/L of sugar, and the average ethanol production was 3.22% *v/v*. The dynamic population of *S. cerevisiae* in T1 presented an increase in the number of cells from $2 \times 10^6$ cells/mL to $1.82 \times 10^8$ cells/mL on day 7, whereas the maximum population achieved by BSc114 (TC, control) was $1.9 \times 10^8$ cells/mL on day 6.

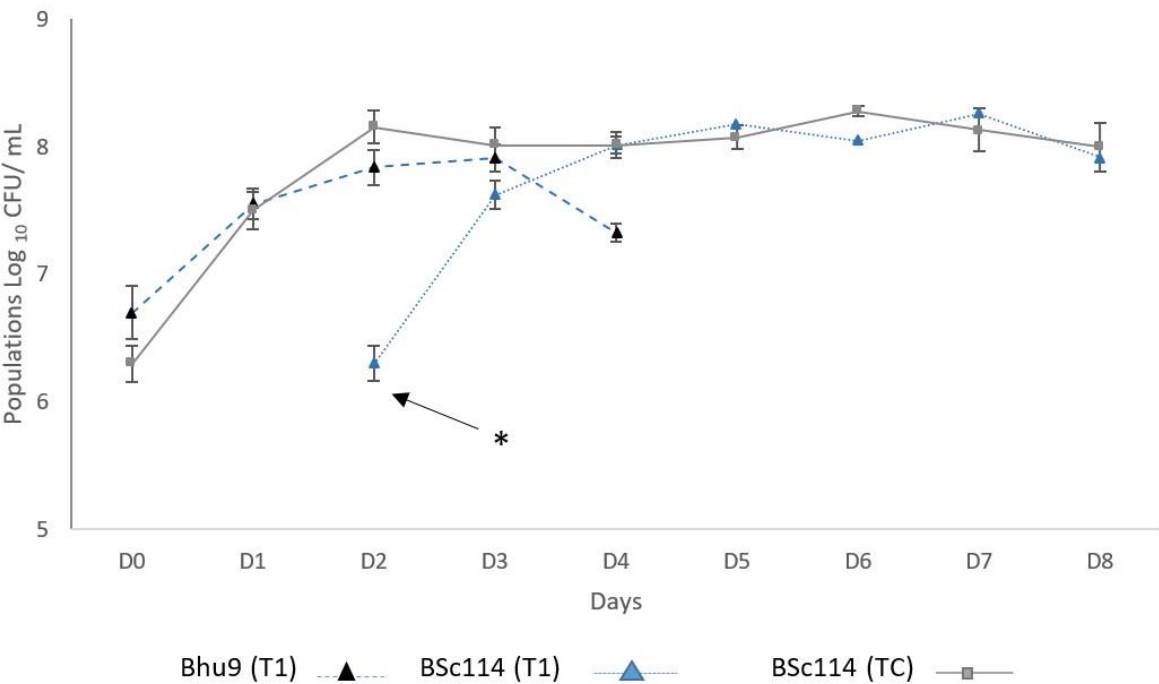

**Figure 2.** Dynamic populations of T1 (BHu9/BSc114) and TC (BSc114). * indicate the inoculation moment of BSc114 in T1 after 48 h.

### 3.2. Enological Parameters

The analyses of the main chemical parameters at the end of the fermentations are summarized in Table 1. Both treatments finished with sugar concentrations below 1.8 g/L, indicating that the fermentations had been successfully finished. Ethanol concentration in T1 was significantly lower (12.63 ± 0.05% *v/v*) than TC (13.15 ± 0.28% *v/v*), representing an average reduction of 0.52% *v/v*. Likewise, pH values were lowest in wines produced with co-cultures, but tartaric acid was higher compared to control (TC) wines. No significant differences were observed for acetic, malic and lactic acid, or for glycerol and total acidity under the experimental conditions.

**Table 1.** Principal chemical parameters in wines obtained from BHu9/Bcs114 co-inoculation and BSc114 control.

| Chemical Compounds | T1 | TC |
|---|---|---|
| Ethanol (%*v/v*) | 12.63 ± 0.05 | 13.5 ± 0.28 (*) |
| Acetic acid (g/L) | 0.56 ± 0.02 | 0.49 ± 0.04 |
| Lactic acid (g/L) | 0.5 ± 0.01 | 0.47 ± 0.02 |
| Malic acid (g/L) | 2.85 ± 0.21 | 2.85 ± 0.07 |
| Tartaric acid (g/L) | 1.31 ± 0.08 | 1 ± 0.01 (*) |
| Glycerol (g/L) | 10 ± 0.71 | 9.3 ± 0.57 |
| pH | 3.43 ± 0.00 | 3.49 ± 0.01 (*) |
| Total acidity (g/L) | 5.95 ± 0.07 | 5.80 ± 0.14 |
| Residual sugar (g/L) | 1.55 ± 0.64 | 1.70 ± 0.28 |

REFERENCES: (*) indicate significant differences between treatments at $p < 0.05$.

### 3.3. Aromatic Composition

Volatile products of the fermented musts were quantified by SPE-GC-MS according to Boido et al. (2003). Table 2 shows volatile compounds and their concentrations, odorant descriptors, perception thresholds, odorant activity values (OAVs), and aromatic series found in the Malbec wines analyzed.

A total of 38 volatile compounds were identified and quantified, and classified into four groups: esters (ethyl and acetate esters), higher alcohols, fatty acids, and lactones.

**Table 2.** Aromatic description of wines obtained (mean and SD of two measure).

| Compounds (µg/L) | Treatments T1 | TC | $p < 0.05$ | Descriptor | Threshold Perception (µg/L) | Ref | OAV T1 | OAV TC | Aromatic Serie |
|---|---|---|---|---|---|---|---|---|---|
| **Ethyl esters** | | | | | | | | | |
| Ethyl hexanoate | 600 ± 15 | 200 ± 83 | * | fruity, apple | 14 | 1 | 42.85 | 14.28 | 5 |
| Ethyl octanoate | 872 ± 58 | 369 ± 187 | * | pineapple, pear | 5 | 1 | 174.4 | 73.8 | 5 |
| Ethyl-3-hydroxybutanoate | 212 ± 53 | 174 ± 14 | | grape, caramel | 67,000 | 2 | 0.003 | 0.002 | 2 |
| Ethyl decanoate | 1140 ± 40 | 590 ± 81 | * | floral | 200 | 1 | 5.7 | 2.95 | 4 |
| Ethyl dodecanoate | 672 ± 52 | nd | | leaf, fruity | 1500 | 1 | 0.448 | - | 5 |
| Ethyl tetradecanoate | 31 ± 1 | 121 ± 22 | * | waxy | 2000 | 1 | 0.015 | 0.060 | 6 |
| Ethyl palmitate | 345 ± 23 | 224 ± 14 | * | waxy | 1500 | 1 | 0.23 | 0.149 | 6 |
| Ethyl succinate | 127 ± 0.011 | 145 ± 6 | * | ripe melon | 1,000,000 | 1 | 0.0001 | 0.0001 | 5 |
| Ethyl lactate | 603 ± 21 | 409 ± 63 | * | strawberrry | 14,000 | 1 | 0.043 | 0.029 | 5 |
| **∑ Ethyl esters** | **4600** | **2232** | | | | | | | |
| **Acetate esters** | | | | | | | | | |
| Isoamyl acetate | 3210 ± 18 | 2892 ± 191 | * | banana | 30 | 2 | 107 | 96.4 | 5 |
| Hexyl acetate | 24 ± 24 | 297 ± 11 | * | red fruit | 1500 | 1 | 0.03 | 0.443 | 5 |
| **∑ acetate esters** | **3234** | **3189** | | | | | | | |
| **TOTAL ESTERS** | **7491 (1.77%)** | **5421 (1.03%)** | | | | | | | |
| **Higher Alcohols** | | | | | | | | | |
| 2-Methyl-1-propanol | 18,480 ± 1620 | 12,990 ± 299 | * | solvent | 7000 | 3 | 2.64 | 1.85 | 1 |
| 1-Butanol | 444 ± 1 | 692 ± 72 | * | solvent | 9000 | 3 | 0.049 | 0.076 | 1 |
| 3-Methyl-1-butanol | 318,400 ± 12,300 | 371,000 ± 16,140 | * | burned, alcohol | 30,000 | 3 | 10.61 | 12.36 | 1 |
| 1-Pentanol | 40 ± 2 | nd | | fruity, balsmic | 4000 | 3 | 0.01 | - | 5 |
| 4-Methyl-1-pentanol | 59 ± 8 | 60 ± 18 | | almond | 50,000 | 3 | 0.001 | 0.001 | 2 |
| 3-Methyl-1-pentanol | 214 ± 18 | 220 ± 10 | | herbaceous | 50,000 | 3 | 0.004 | 0.044 | 3 |
| 1-Hexanol | 1080 ± 142 | 852 ± 82 | | grass, green leaf | 2500 | 4 | 0.432 | 0.340 | 3 |
| *trans*-3-Hexenol | 40 ± 0 | nd | | herbaceous, land | 400 | 1 | 0.1 | - | 3 |
| 3-Ethoxy-1-propanol | 150 ± 11 | 65 ± 3 | * | ripe pear | 100 | 1 | 1.5 | 0.65 | 5 |
| *cis*-3-Hexenol | 54 ± 8 | 57 ± 8 | | cutted grass | 400 | 1 | 0.135 | 0.014 | 6 |
| 2-Ethyl hexanol | 256 ± 12 | 255 ± 17 | | rose, citrus | 8000 | 1 | 0.032 | 0.031 | 4 |
| 2,3-Butanediol | 233 ± 15 | 340 ± 13 | * | butter | 120,000 | 1 | 0.001 | 0.002 | 6 |
| Furfurol | 102 ± 19 | 190 ± 20 | * | floral | 5000 | 4 | 0.02 | 0.038 | 4 |
| 3-(Methylthio)-1-propanol | 819 ± 41 | 1450 ± 38 | * | cooked vegetal | 1000 | 5 | 0.819 | 1.45 | 3 |
| Benzyl alcohol | 610 ± 50 | 11 ± 4 | * | caramelo, cítrico | 10,000 | 4 | 0.061 | 0.0001 | 2 |
| 2-Phenylethyl alcohol | 53,885 ± 3012 | 86,072 ± 731 | * | rose | 14,000 | 4 | 3.848 | 6.148 | 2 |
| Tyrosol | 17,110 ± 895 | 23,071 ± 3245 | * | honey | - | - | | | 2 |
| Tryptophol | 1910 ± 98 | 1780 ± 98 | * | | - | - | | | |

**Table 2.** *Cont.*

| Compounds (µg/L) | Treatments T1 | TC | p < 0.05 | Descriptor | Threshold Perception (µg/L) | Ref | OAV T1 | OAV TC | Aromatic Serie |
|---|---|---|---|---|---|---|---|---|---|
| Σ Higher alcohols | 414,357 (98.14%) | 515,754 (98.45%) | | | | | | | |
| **Fatty acids** | | | | | | | | | |
| Acetic acid | 39 ± 6 | nd | | vinegar | 200 | 1 | 0.195 | - | 6 |
| Isobutanoic acid | 181 ± 21 | 396 ± 9 | * | butter, cheese | 8100 | 3 | 0.022 | 0.048 | 6 |
| Butanoic acid | 56 ± 3 | 295 ± 29 | * | fatty, rancid | 1000 | 3 | 0.056 | 0.295 | 6 |
| Hexanoic acid | 176 ± 53 | 231 ± 42 | * | cheese, sudor | 3000 | 3 | 0.058 | 0.077 | 6 |
| Octanoic acid | 235 ± 50 | 824 ± 9 | * | rancid butter | 3000 | 3 | 0.078 | 0.276 | 6 |
| Decanoic acid | 94 ± 9 | 773 ± 310 | * | fatty, rancid | 10,000 | 3 | 0.009 | 0.007 | 6 |
| Dodecanoic acid | 49 ± 1 | 43 ± 1 | | fatty, rancid | 10,000 | 3 | 0.005 | 0.004 | 6 |
| Σ Acids | 83 (0.019%) | 2562 (0.49%) | | | | | | | |
| **Lactones** | | | | | | | | | |
| gamma-Valerolactone | 39 ± 1 | 49 ± 3 | * | sweet, cocconut | 10 | 7 | 3.9 | 4.9 | 2 |
| gamma-Butyrolactone | 236 ± 8 | 163 ± 26 | * | caramel | 35 | 7 | 6.74 | 4.65 | 2 |
| Σ Lactones | 275 (0.06%) | 212 (0.04%) | | | | | | | |
| Σ compounds (µg/L) | 422,206 | 523,858 | | | | | | | |

References: (*) indicate significant differences between treatments. Grey cells indicate compounds with OAV > 1. References: [1] Tao and Zang (2010), [2] Moreno et al. (2005), [3] Rychlik et al. (1996), [4] Leffingwell & Associates (2009), [5] Burdock (2016), [6] Culleré et al. (2004), [7] Lopez de Lerma and Peinado (2011).

Alcohols formed the most abundant group of volatile compounds, followed by esters, fatty acids and lactones. Higher alcohols represented 98.14 and 98.45% of the total aroma content in T1 and TC wines, respectively, while esters and fatty acids constituted 1.77–1.03% and 0.019–0.49% in T1 and TC wines, respectively (Table 2).

Overall, the total concentration of higher alcohols and fatty acids was higher in control treatment TC, fermented with *S. cerevisiae* BSc114, than in wines produced by the sequential fermentation of *H. uvarum* BHu9/*S. cerevisiae* BSc114 (T1). In contrast, esters and lactones (γ-butyrolactone and γ-valerolactone) content was higher in T1 than in TC. These compounds represented 1.77 and 1.03 % in esters, in T1 and TC respectively. The lactones proportions were 0.06 % and 0.04 % in T1 and TC respectively. (Table 2).

Some compounds such as ethyl hexanoate, ethyl decanoate, 3-ethoxy-1-propanol, isoamyl acetate, and γ-butyrolactone were detected at higher concentrations in T1 than in TC ($p < 0.05$). In contrast, *S. cerevisiae* BSc114 fermentation showed higher concentrations of ethyl octanoate, 3-methyl-1-butanol, 3-(methyl thio)-1-propanol, 2-phenylethanol, and γ-valerolactone compared to wines obtained with BHu9/BSc114 ($p < 0.05$) (Table 2).

Compounds such us ethyl hexanoate (fruity, apple), ethyl octanoate (pineapple, pear), isoamyl acetate (banana), γ-butyrolactone (caramel, coconut), and γ-valerolactone (sweet, coconut) showed OAVs > 1 in both treatments. Comparing pure with mixed, fermented 3-ethoxy-1-propanol (ripe pear) exhibited an OAV > 1 only in T1, and 3-(methyl thio)-1-propanol (cooked vegetables) showed an OAV > 1 in wine fermented by the *S. cerevisiae* Bc114 monoculture (Table 2).

Table 3 presents compounds with an OAV > 1 and their relative odor contribution (ROC). When considering the ester contribution to the odorant composition, ethyl octanoate greatly contributed to wines obtained with BHu9/BSc114 (49.35%), while isoamyl acetate was the main contributor to the control treatment fermented with pure BSc114 control. The higher alcohols that demonstrated major contributions to wines in both treatments were 3-methyl-1-butanol and 2-phenylethanol, but their relative odor contributions were higher in TC wines.

**Table 3.** Compounds with an OAV > 1 and their relative odor contribution (ROC) in T1 and TC wines.

| Compounds | T1 | | TC | | Aromatic Serie |
|---|---|---|---|---|---|
| | OAV | ROC (%) | OAV | ROC (%) | |
| Ethyl hexanoate | 42.85 | 12.12 | 14.28 | 6.60 | 5 Fruity |
| Ethyl octanoate | 174.40 | 49.35 | 73.8 | 34.09 | 5 Fruity |
| Ethyl decano ate | 1.03 | 0.29 | 0 | 0.00 | 4 Floral |
| Isoamyl acetate | 107.00 | 30.28 | 96.40 | 44.53 | 5 Fruity |
| 2-Methyl-1-propanol | 2.64 | 3.72 | 1.85 | 6.51 | 1 Solvent |
| 3-Methyl-1-butanol | 10.61 | 3.00 | 12.36 | 5.71 | 1 Solvent |
| 3-Ethoxy-1-propanol | 1.50 | 0.42 | <1 | - | 5 Fruity |
| 3-(Methylthio)-1-propanol | <1 | - | 1.45 | 0.67 | 3 Herbaceous |
| 2-Phenylethanol | 5.38 | 1.52 | 8.61 | 3.98 | 4 Floral |
| gamma-Valerolactone | 3.90 | 1.10 | 4.90 | 2.26 | 2 Sweet |
| gamma-Butirolactone | 6.742 | 1.91 | 4.66 | 2.15 | 2 Sweet |

Figure 3 shows the aromatic profile of the analyzed wines based on the sum of the components with an OAV > 1 and ROC values according to each descriptor. Wines fermented with BSc114 were related to the aromatic "floral", "solvent", "herbaceous" and "sweet" families, while co-culture fermented wines were characterized by "frutal" descriptors.

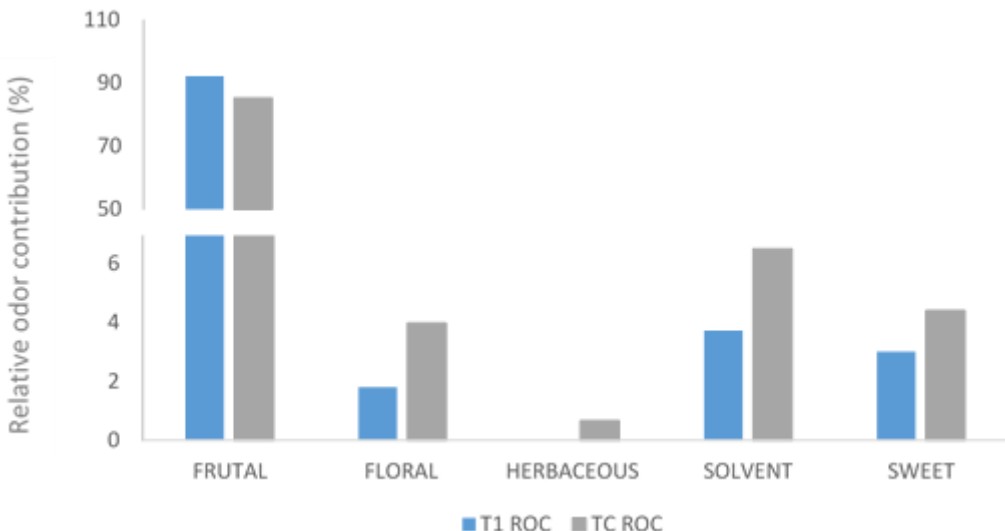

**Figure 3.** Aromatic profile of wines produced by BHu9/BSc114 (T1) and BSc114 (TC control).

*3.4. Sensorial Analysis*

Figure 4 shows the sensorial analyses of the wines obtained. Wines fermented with *H. uvarum* BHu9/*S. cerevisiae* BSc114 (T1) could be defined as fruity ($p < 0.05$). These results are in agreement with the aromatic profile obtained with ROC analysis. Another parameter that significantly affected reduced ethanol wines (T1) was color intensity. Wines produced with *S. cerevisiae* BSc114 (TC) were more related to the floral descriptor, which is in agreement with the ROC results due to 2-phenylethanol concentration; moreover, these wines were associated with astringency and hotness mouthfeel.

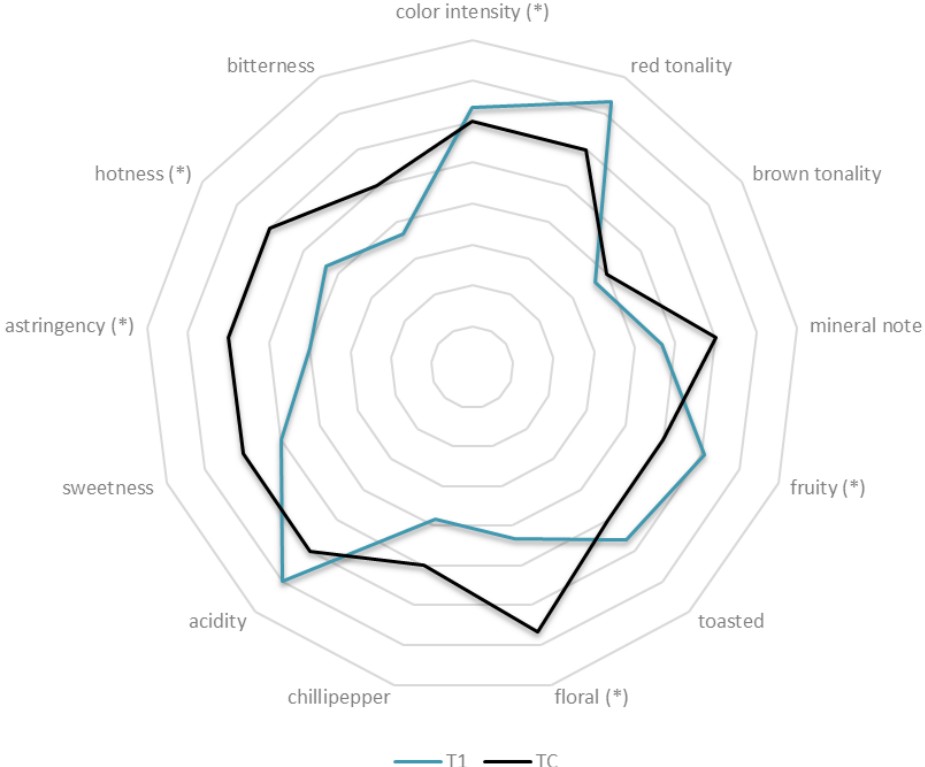

**Figure 4.** Sensory analysis of wines obtained from BHu9/BSc114 (T1) and BSc114 (TC). (*) Difference significant at 95% confidence level.

## 4. Discussion

The current wine market requires wines with lower ethanol concentrations and complex flavor and color perception. Several microbiological strategies have been proposed in order to obtain these characteristics. The present study intends to verify the behavior of selected yeasts regarding ethanol production and the sensorial impact on the wine quality.

Several studies have reported reduced ethanol levels with sequential inoculations of non-*Saccharomyces* and *Saccharomyces* yeasts and under different winemaking conditions [4,16,31]. It is well known that *H. uvarum* is the most representative yeast species found on grape surfaces showing prevalence during early stages of spontaneous alcoholic fermentation [32]. This yeast has several characteristics that could be used to reduce ethanol content in wines [21]. In the current study, inoculation of *H. uvarum* BHu9 prior to inoculation of *S. cerevisiae* BSc114 demonstrated a sugar consumption of 35.7 g/L for 1% of ethanol produced. In contrast, *S. cerevisiae* control (TC) used 17.5 g /L of sugar for 1% of ethanol produced. It is reported that *S. cerevisiae* yeast uses 16.83 to 17 g/L on average [1]. The decrease in ethanol production can sometimes be explained by an increase in glycerol and acetic acid. However, in the present study, both glycerol and acetic acid did not show significant differences. Sugars were probably partially consumed through the oxidative pathway to produce biomass and other products.

It must be highlighted that when both strains remained together, sugar consumption was lower than in the *H. uvarum* monoculture (prior to *S. cerevisiae* inoculation). There is evidence that presence of non-*Saccharomyces* yeasts in the early stages of fermentation could affect the metabolic activity of *S. cerevisiae*, probably encouraging a competition for nutrients [33–35]. For example, Bisson et al. [36] demonstrated that *K. apiculata* consumed thiamine and other micronutrients, generating inefficiency in the metabolic development of *S. cerevisiae*. Another study established that immobilized *Starmerella bombicola* cells in a mixed fermentation affected decarboxylase and alcohol dehydrogenase levels of *S. cerevisiae* [37]. Recently, the research of Petitgonnet et al. [38] demonstrated that sequential culture between *Lachaceae thermotholerans* and *S. cerevisiae* provokes a negative interaction between the two species to the detriment of *S. cerevisiae*, due to a cell–cell contact mechanism and essential nutrients uptake. When *H. uvarum* and *S. cerevisiae* are mixed inoculated, the cultivability of *H. uvarum* is significantly affected; however, the final ethanol concentrations are lower compared to the pure culture of *S. cerevisiae* [39]. Nevertheless, to answer the results of this work, further studies should be carried out to fully understand the interactions between *H. uvarum* and *S. cerevisiae* strains employed in the present study.

With respect to the aromatic composition, many studies have shown that non-*Saccharomyces* yeasts such as *Candida, Debaryomyces, Pichia, Hansenula,* and *Hanseniaspora,* that display oxidative metabolism and/or are weakly fermentative produced higher ester levels than a single *S. cerevisiae* culture [40]. In accordance, the total ester concentrations in wines produced by *H. uvarum/S. cerevisiae* co-cultures were superior to that of wines produced by control treatment. The co-inoculation showed higher levels of ethyl hexanoate, ethyl octanoate and ethyl decanoate which allowed the "fruity" aromatic profile of the wines.

Fusel alcohol production was higher in wine fermented with a monoculture of *S. cerevisiae*. Fusel alcohol production is related to amino acid production by yeasts, which varies according to genera, species and strain [41]. *S. cerevisiae* yeasts have been reported to produce higher quantities of these compounds compared with certain non-*Saccharomyces* yeasts [42]. The aromatic series that best describes the TC profile is "floral", which is associated with 2-phenylethanol levels. As was expected, sensorial analysis of TC wine related it to floral descriptors ($p < 0.05$).

It is well known that ethanol significantly affects the sensorial perception of wines [43]; for example, it decreases the perception of higher alcohols and aldehydes and shows a similar effect for ethyl esters [44]. According to our results, wines obtained from a BSc114 (TC) monoculture could be associated with attributes such us astringency, bitterness, hotness, and sweetness, which is in agreement with the results by Tilloy et al. [45]. The authors found that high ethanol levels enhanced

the perception of the abovementioned attributes. In contrast, wines with lower ethanol levels obtained with *H. uvarum/S. cerevisiae* in the present study were related to red fruit by the sensorial panel ($p < 0.05$). Although the control treatment presented elevated concentrations of chemical compounds which are commonly related to fruity descriptors, it has been reported that high ethanol contents can mask certain flavor-related volatile compounds like those related to fruity and floral profiles [46].

To our knowledge, this is the first time that a *H. uvarum* species, submitted to a previous selection process, has been proposed to carry out sequential fermentations with *S. cerevisiae* under optimized conditions to reduce ethanol in wines. The results obtained in the present study have demonstrated the impact of this co-culture on the ethanol concentration and the chemical aromatic composition; and, in addition, it has evidenced that ethanol levels affect sensorial perception. Therefore, the present study could be considered an additional step to a successful change in the wine industry to face current consumers' demands. It is possible, however, that more research is necessary in order to fully understand the impact of this co-culture on a major production scale.

**Author Contributions:** Investigation, M.V.M.; Methodology, M.V.M., Y.P.M., C.G. and E.D.; Project administration, Y.P.M., M.E.T., F.V. and E.D.; Supervision, F.C.; Writing—original draft, M.V.M., F.V. and E.D.; Writing—review & editing, M.V.M., Y.P.M., M.C., L.M., M.E.T. and F.C.

**Funding:** This research was funded by [UNSJ-SECITI] grant number [3635-2015-2017] and [UNSJ-CICITCA] grant number [1531-2016].

**Conflicts of Interest:** The authors declare no conflict of interest.

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
