# Peer review of "Impact on Sensory and Aromatic Profile of Low Ethanol Malbec Wines Fermented by Sequential Culture of Hanseniaspora uvarum and Saccharomyces cerevisiae Native Yeasts"

_fermentation, doi:10.3390/fermentation5030065_

Reviewer 1 Report

Comments and Suggestions for Authors

The present manuscript describes a microbiological strategy for reducing ethanol content in wines and study its aromatic and sensorial impact in order to answer the current consumers’ preferences and address the main challenges faced by the oenological industry. In my opinion, this study is well written and structured and has great relevance to wine sector, considering that currently there is an increasing interest for non-Saccharomyces yeasts to be used in winemaking. Therefore, this manuscript should be published after minor revision.

Specific recommendations

In general, the genera and species names should be checked by authors and appear in italics, as well as, the subscripts or superscripts must be corrected.

In the section 2.8. Odor activity value (OAV) and relative odor contributions (ROCs), this part is a bit convoluted, please reorder the paragraphs. Moreover, a short explanation of ROC concept will be necessary. In relation to ROC determination, ROC has been calculated by authors as individual OAV/∑OAV or the ratio between individual OAV and total OAV that showed OAV>1 as estimated by Welke et al.

In Table 3, please add the compound 2-methyl-1-propanol since its OAV value is >1 in both wines studied. Therefore, their OAV values must be highlighted in grey on Table 2.

Figure 3 should be modified by authors, including the relative odor contribution of 2-methyl-1-propanol as part of “solvent”.

Page 2, line 44: please correct in [1].

Page 2, line 57: please correct in [17, 18].

Page 2, line 86: please delete “In order to reduce the lag-stage in the grape must” because this information is repeated again in the same paragraph (page 3, lines 89-90).

Page 3, line 105: please use the abbreviation H. uvarum

Page 7, line 223, Table 1: please add the significance value at the foot of the table.

Page 8, Table 2: please change to “pineapple”

Page 9, Table 2: ∑compounds (mg/L)…Correct in: ∑compounds (µg/L)

Page 10, lines 241-243: These two sentences repeat the same results, please summarize in a single sentence.

Page 10, lines 253-254: please review this conclusion “Comparing pure with….exhibited an OAV>1 only in T1”……….. because the OAV of ethyl decanoate is >1 in T1 and TC wines.

Page 12, Figure 4: please change “biterness” to “bitterness”

Page 13, lines 317-329: please add some studies where the presence of H. uvarum have influenced the S. cerevisiae growth.

Page 13, line 325: please change “Lanchaceae thermotholerans” to “Lachancea thermotolerans

Page 13, line 333: please add references.

Page 14, line 359: please delete “Is possible”

Author Response

Dear editor

I appreciate the corrections made by reviwer 1

In red are remarked reviwer suggestions 

In yellow I respond to the suggestions made by reviewer 1

In manuscript is remarked with yellow color the corrections made

Specific recommendations

In general, the genera and species names should be checked by authors and appear in italics, as well as, the subscripts or superscripts must be corrected.

Response: genera and species names were corrected in italics format

In the section 2.8. Odor activity value (OAV) and relative odor contributions (ROCs), this part is a bit convoluted, please reorder the paragraphs. Moreover, a short explanation of ROC concept will be necessary. In relation to ROC determination, ROC has been calculated by authors as individual OAV/∑OAV or the ratio between individual OAV and total OAV that showed OAV>1 as estimated by Welke et al. 

Response:  The sentences corresponding at section 2.8 was improved

In Table 3, please add the compound 2-methyl-1-propanol since its OAV value is >1 in both wines studied. Therefore, their OAV values must be highlighted in grey on Table 2.

Response: compound 2-methyl-1-propanol were hidhlighted in grey color on table 2 and included in table 3

Figure 3 should be modified by authors, including the relative odor contribution of 2-methyl-1-propanol as part of “solvent”.

Response: 2 Methyl 1 propanol was included in contributions in figure 3 and Table 3

Page 2, line 44: please correct in [1].

Response: wsa corrected

Page 2, line 57: please correct in [17, 18].

Response: parenthesis was eliminated 

Page 2, line 86: please delete “In order to reduce the lag-stage in the grape must” because this information is repeated again in the same paragraph (page 3, lines 89-90).

Response: “In order to reduce the lag-stage in the grape must” was eliminated, and Line 86 was improved

Page 3, line 105: please use the abbreviation H. uvarum

Response: the abbreviation was made 

Page 7, line 223, Table 1: please add the significance value at the foot of the table.

Response: The significance value was placed at the foot of the table.

Page 8, Table 2: please change to “pineapple”

Response: peneapple was changed to pineapple

Page 9, Table 2: ∑compounds (mg/L)…Correct in: ∑compounds (µg/L)

Response: the change was made

Page 10, lines 241-243: These two sentences repeat the same results, please summarize in a single sentence.

 Response : the sentence "In addition, wines produced from mixed fermentations showed higher amounts of esters and lactones compared to pure S. cerevisiae control fermentations. "  was eliminated due to repeat the same results that previous sentence

Page 10, lines 253-254: please review this conclusion “Comparing pure with….exhibited an OAV>1 only in T1”……….. because the OAV of ethyl decanoate is >1 in T1 and TC wines.

Response: ethyl decanoate was eliminated of the sentence

Page 12, Figure 4: please change “biterness” to “bitterness”

Response:  was corrected

Page 13, lines 317-329: please add some studies where the presence of H. uvarum have influenced the S. cerevisiae growth.

Response: reference respect to H. uvarum/S. cerevisiae interaction was included 

39 Wang, C., Mas, A., & Esteve-Zarzoso, B. (2015). Interaction between Hanseniaspora uvarum and Saccharomyces cerevisiae during alcoholic fermentation. Int J food microbiol, 206, 67-74.

Page 13, line 325: please change “Lanchaceae thermotholerans” to “Lachancea thermotolerans

Response: the name was changed

Page 13, line 333: please add references.

Response: the reference was included

40 Erten, H., & Campbell, I. (1953). The production of low‐alcohol wines by aerobic yeasts. J I Brewing, 59(3), 207-215.

Page 14, line 359: please delete “Is possible”

Response: "Is possible) was eliminated

Reviewer 2 Report

The manuscript dealt with the exploitation of sequential fermentation with H. uvarum and S. cerevisiae for the production of low ethanol wine. The paper sound well and provided a complete description of fermentative kinetics and aroma profiles of wine samples resulted from two trials, namely one with Sc alone (called TC) and another with H. uvarum + Sc (called T1). However, several other papers described the use of non-conventional yeasts in winemaking to reduce ethanol content and increase aroma complexity. The authors should better explain what is new in their study compared to the previous ones. For example, is the ethanol reduction obtained in this work comparable with the ethanol decrease obtained in other paper? Another drawback is the lack of fermentative trial with H. uvarum alone. This assay should give a significant contribution to understand how Sc and H. uvarum interacted each other to improve wine aroma complexity and reduce ethanol content. Finally, some references are missing, i.e. Hu et al. Food Research International, 2018; Gobbi et al. European Food Research and Technology 2014, Volume 239, Issue 1, pp 41–48 (this list is not exhaustive).

Minor Remarks

55-56. Genus and species names in italics. Please check it through the text.

88. Why you supplemented the must with YE and peptone?

90-91. This statement is redundant. The same information was reported above.

100. What did you use to fill the Muller valve?

101. Why S. cerevisiae was inoculated 24h later than H. uvarum? What is the advantage of sequential fermentation in comparison to co-colture?

217. Was the ethanol difference between T1 and TC statistically significant?

242-245 This part is unclear. Please re-write it.

Author Response

Please, find you bellow the requests to the sugestions to reviewer 2

# Reviewer 2

The manuscript dealt with the exploitation of sequential fermentation with H. uvarum and S. cerevisiae for the production of low ethanol wine. The paper sound well and provided a complete description of fermentative kinetics and aroma profiles of wine samples resulted from two trials, namely one with Sc alone (called TC) and another with H. uvarum + Sc (called T1). However, several other papers described the use of non-conventional yeasts in winemaking to reduce ethanol content and increase aroma complexity. The authors should better explain what is new in their study compared to the previous ones.

We agree with the Reviewer 2, regarding that several investigations describe the use of NS in order to reduce ethanol concentrations in wines. Nevertheless, our research group has been working in selection NS yeasts (Mestre et al, 2017), later these selected yeasts were subjected to optimization process in order to perform sequential inoculations that it allow reduce ethanol in wines. In the last instance (in the present study) we demonstrate that optimized strategy positively affects the aromatic and sensorial profile in wine. The results obtained in the present study have demonstrated the impact of this co-culture on the ethanol concentration and the chemical aromatic composition and in addition, it has evidenced that ethanol levels affect the sensorial perception. Therefore, we consider that the present study is the closing of a series of works (Mestre et al., 2017, Maturano et al., 2019) which can be transferred as successful tool at the wine industry to face current consumers’ demands.

For example, is the ethanol reduction obtained in this work comparable with the ethanol decrease obtained in other paper?

The reduction ethanol in this work (0.52%v/v) is significant and comparable to others works such as:

-Englezos et al. 2019 (https://doi.org/10.1016/j.foodres2019.03.072): significant average reduction of 0.55% v/v

- Englezos el al. 2016 (htpp//doi.org/10.1007/s00253-016-7413-z): significant average reduction of 0.5% v/v

- Tilloy et al 2014 (https://doi:10.1128/AEM.03710-13)  reported ethanol level reductions of 0.45% (v/v) in Grenache and 0.61% (v/v) in Syrah

Another drawback is the lack of fermentative trial with H. uvarum alone. This assay should give a significant contribution to understand how Sc and H. uvarum interacted each other to improve wine aroma complexity and reduce ethanol content.

We agree with the reviewer that the fermentations with H. uvarum monoculture would have provided information on its metabolism and behavior and its impact on the complexity of wine, however, the yeasts behave very differently when it ferment alone and when they are under co-culture conditions. Moreover, when we design the experiment considered that it was not necessary to carry out fermentations with the yeast non- Saccharomyces alone since in the industry it is never done. NS Yeasts cannot finish a fermentation.

Finally, some references are missing, i.e. Hu et al. Food Research International, 2018; Gobbi et al. European Food Research and Technology 2014, Volume 239, Issue 1, pp 41–48 (this list is not exhaustive)

We don’t understand you because these references are not in the manuscript or in the list of references

We do not consider it necessary to include these references in the manuscript.

Minor Remarks

55-56. Genus and species names in italics. Please check it through the text.

Names were re-write in italic format.

88. Why you supplemented the must with YE and peptone?

In yeast inoculum preparation, we added YE and Peptone at grape must in order to improve the nutritional composition of the grape must. 

90-91. This statement is redundant. The same information was reported above.

The paragraph corresponding to section 2.2 Yeast inoculation was modified in order to avoid the sentences duplicated. Line 85-92.

100. What did you use to fill the Muller valve?

The Muller valve were filled with a solution of 50% of sulfuric acid and 50% sterile water distilled. We add this specification in the manuscript. Lines 101-102.

101. Why S. cerevisiae was inoculated 24h later than H. uvarum? What is the advantage of sequential fermentation in comparison to co-colture?

In this study S. cerevisiae was inoculated 48 h later to H. uvarum inoculations (line 107). This inoculation strategy was previously optimized by our research group, Maturano et al., 2019 (Maturano, Y. P., Mestre, M. V., Kuchen, B., Toro, M. E., Mercado, L. A., Vazquez, F., and Combina, M. (2019). Optimization of fermentation-relevant factors: A strategy to reduce ethanol in red wine by sequential culture of native yeasts. Int J Food Microbiol, 289, 40-48). We reported that S. cerevisiae inoculations 48 h later H. uvarum reduce significantly ethanol concentrations in wines. In the present study we want to demonstrate that the optimized inoculation strategy positively affects the aromatic and sensory profile of the wines.

Sequential culture allows a better metabolic behavior of NS yeast strains. We consider that is crucial a good implantation and persistence of NS in fermentative process. For this reason, we propose this type inoculation.

217. Was the ethanol difference between T1 and TC statistically significant?

The ethanol concentration was statistically significant (p value < 0.05). This date was placed at the foot of the table 1.

242-245 This part is unclear. Please re-write it.

The sentence was modified, because there was a repetition of results.

In contrast, esters and lactones (γ-butyrolactone and γ-valerolactone) content was higher in T1 than in TC. These compounds represented 1.77 and 1.03 % in esters, in T1 and TC respectively. While 0.06 and 0.04% of lactones proportions respect to total aromatic matrix analyzed in both treatments (Table 2). (Lines 244-246).